# EVALUATIONS AND METHODS FOR EXPLANATION THROUGH ROBUSTNESS ANALYSIS

## ABSTRACT

Among multiple ways of interpreting a machine learning model, measuring the importance of a set of features tied to a prediction is probably one of the most intuitive way to explain a model. In this paper, we establish the link between a set of features to a prediction with a new evaluation criterion, robustness analysis, which measures the minimum distortion distance of adversarial perturbation. By measuring the tolerance level for an adversarial attack, we can extract a set of features that provides the most robust support for a current prediction, and also can extract a set of features that contrasts the current prediction to a target class by setting a targeted adversarial attack. By applying this methodology to various prediction tasks across multiple domains, we observe the derived explanations are indeed capturing the significant feature set qualitatively and quantitatively.

## 1 INTRODUCTION

With the significant progress of recent machine learning research, various machine learning models have been being rapidly adopted to countless real-world applications. This rapid adaptation increasingly questions the machine learning model's credibility, fairness, and more generally *interpretability*. In the line of this research, researchers have explored various notions of model interpretability. Some researchers directly answer the trustability (Ribeiro et al., 2016) or the fairness of a model (Zhao et al., 2017), while some other researchers seek to actually improve the model's performance by understanding the model's weak points (Koh & Liang, 2017). Even though the goal of such various model interpretability tasks varies, vast majority of them are built upon extracting relevant features for a prediction, so called feature-based explanation.

Feature-based explanation is commonly based on measuring the fidelity of the explanation to the model, which is essentially how close the sum of attribution scores for a set of features approximates the function value difference before and after removing the set of features. Depending on their design, the fidelity-based attribution evaluation varies: completeness (Sundararajan et al., 2017), sensitivity-n (Ancona et al., 2018), infidelity (Yeh et al., 2019), and causal local explanation metric (Plumb et al., 2018). The idea of smallest sufficient region (SSR) and smallest destroying region (SDR) (Fong & Vedaldi, 2017; Dabkowski & Gal, 2017) is worth noting because it considers the ranking of the feature attribution scores, not the actual score itself. Intuitively, for a faithful attribution score, removing the most salient features would naturally lead to a large difference in prediction score. Therefore, SDR-based evaluations measure how much the function value changes when the most high-valued salient features are removed.

Although the aforementioned attribution evaluations made success in many cases, setting features with an arbitrary reference values to zero-out the input is limited, in the sense that it only considers the prediction at the reference value while ignoring the rest of the input space. Furthermore, the choice of reference value inherently introduces bias. For example, if we set the feature value to 0 in rgb images, this introduces a bias in the attribution map that favors the bright pixels. As a result, explanations that optimize upon such evaluations often omit important dark objects and the *pertinent negative* features in the image, which is the part of the image that does not contain object but is crucial to the prediction (Dhurandhar et al., 2018). An alternative way to remove pixels is to use sampling from some predefined distribution or a generative model (Chang et al., 2018), which nevertheless could still introduce some bias with respect to the defined distribution. Moreover, they require a generative model that approximates the data distribution, which may not be available in certain domains.

In this paper, we remove such inherit bias by taking a different perspective on the input perturbation. We start from an intuition that if a set of features are *important* to make a specific prediction, keeping them in the same values would preserve the prediction even though other irrelevant features are modified. In other words, the model would be more sensitive on the changes of those *important* or *relevant* features than the ones that are not. Unlike the foremost approaches including SDR and SSR that perturbs features to a specific reference point, we consider the minimum norm of perturbation to arbitrary directions, not just to a reference point, that can change model's prediction, also known as "minimum adversarial perturbation" in the literature (Goodfellow et al., 2014; Weng et al., 2018b).

Based on this idea, we define new evaluation criteria to test the importance of a set of features. By computing the minimum adversarial perturbation on the complementary set of features that can alter the model's decision, we could test the degree of importance of the set. Although explicitly computing the importance value is NP-hard (Katz et al., 2017), Carlini & Wagner (2017) and Madry et al. (2017) showed that the perturbations computed by adversarial attacks can serve as reasonably tight upper bounds, which lead to an efficient approximation for the proposed evaluation.

Furthermore, we can derive a new explanation framework by formulating the model explanation to a two-player min-max game between explanator and adversarial attacker. The explanator aims to find a set of important features to maximize the minimum perturbation computed by the attacker. This framework empirically performs much better than previous approaches quantitatively, with very inspiring examples.

To summarize our contributions:

- We define new evaluation criteria for feature-based explanations based on robustness analysis. The evaluation criteria consider the worst case perturbations when a set of features are anchored, which does not introduce bias into the evaluation.
- We design efficient algorithms to generate explanations that maximize the proposed criteria, which perform favorably against baseline methods on the proposed evaluation criteria.
- Experiments in computer vision and NLP models demonstrate that the proposed explanation can indeed identify some important features that are not captured by previous methods. Furthermore, our method is able to extract a set of features that contrasts the current prediction to a target class.

## 2    ROBUSTNESS ANALYSIS FOR EVALUATING FEATURE-BASED EXPLANATIONS

### 2.1    PROBLEM NOTATION

Let us consider the following setting: a general $K$-way classification problem with input space $\mathcal{X} \subseteq \mathbb{R}^d$, output space $\mathcal{Y} = \{1, \ldots, K\}$, and a predictor function $f : \mathcal{X} \to \mathcal{Y}$ where $f(\boldsymbol{x})$ denotes the output class for some input example $\boldsymbol{x} = [\boldsymbol{x}_1, \ldots, \boldsymbol{x}_d] \in \mathcal{X}$. Then, for a particular prediction $f(\boldsymbol{x}) = y$, despite the different forms of existing feature-based explanations ranging from attributing an importance value to each feature, ranking the features by their importance, to simply identify a set of important features, a common goal of them is to extract a compact set of relevant features with respect to the prediction.

### 2.2    EVALUATION THROUGH ROBUSTNESS ANALYSIS

We note that however, given an explanation that identifies a set of *said to be* relevant features, how can we evaluate the quality of such explanation, or in other words, justify whether the distinguished features are truly relevant to the prediction? While one generally has no ground truth about the underlying true relevance of the features, recent studies take an axiomatic approach to define what properties the *relevant* features should hold and evaluate the explanations through verifying if the *identified* relevant features satisfy the properties. One such properties that is widely adopted in the literature is to assume that *the importance of a set of features corresponds to the degree of change in prediction when the features are removed from the original input*. Nevertheless, as we discussed in the previous section, the practice of approximating removal of features by setting their value to some reference point poses the risk of introducing bias in the evaluation. As a result, to escape from the caveat, we follow a similar concept but propose two new criteria to evaluate the importance of features based on the following assumptions.

**Assumption 1**  When the values of the most salient features are anchored (fixed), perturbation on the complementary set of features has weaker influence on the model's prediction. In other words, the model could tolerate a larger degree of perturbation on the less important and non-anchored features.

**Assumption 2**  If perturbation is allowed on a set of important features, a small perturbation could easily change the model prediction even when we fix the values for the rest of the features.

Based on these two assumptions, we propose a new framework for evaluating explanations. The evaluation is based on the adversarial robustness when a set of features are fixed, which is formally defined below.

**Definition 2.1** *The minimum adversarial perturbation norm on a set of features S, which we will also name as Robustness-S, can be defined as:*

$$\epsilon_S^* = g(\boldsymbol{x}, S) = \{\min_{\boldsymbol{\delta}} \|\boldsymbol{\delta}\|_p \text{ s.t. } f(\boldsymbol{x} + \boldsymbol{\delta}) \neq y, \ \boldsymbol{\delta}_{\overline{S}} = 0\}, \tag{1}$$

*where $\overline{S} = U \setminus S$ is the complementary set of features, and $\boldsymbol{\delta}_{\overline{S}} = 0$ means that the perturbation value on features in $\overline{S}$ is constraint to be 0.*

Assume that we are given an explanation that partitions the input features into a relevant set $S_r$ and an irrelevant set $\overline{S_r}$. Assumption 1 implies that the quality of the relevant set can be measured by $\epsilon_{\overline{S_r}}^*$ – the robustness of irrelevant set when the relevant set is anchored. Specifically, Assumption 1 infers that a higher robustness on $\overline{S_r}$ follows from a larger coverage of pertinent features in set $S_r$; and thus an explanation is considered better if it leads to a higher robustness against perturbation in $\overline{S_r}$. On the other hand, based on Assumption 2, an explanation that has included important salient features in $S_r$ should lead to a smaller robustness level on $\epsilon_{S_r}^*$. Therefore, Assumption 1 and 2 build up our proposed evaluation criteria *Robustness-$\overline{S_r}$* and *Robustness-$S_r$* respectively, as listed below.

**Robustness-$\overline{S_r}$**  measures the minimum adversarial distortion $\epsilon_{\overline{S_r}}$ when the set of important features $S_r$, typically represented by the high-weight features in an attribution map, are anchored and perturbation is only allowed in low-weight regions. The higher the score the better the explanation. To measure Robustness-$\overline{S_r}$, we would need to first determine the size of $|S_r|$. We can set $|S_r|$ to the amount of anchors that an user is interested in or we may vary the size of $|S_r|$ and evaluate the corresponding Robustness-$\overline{S_r}$ at different points. By varying the size of $|S_r|$, we could plot an evaluation curve for each explanation and in turn measure the area under curve (AUC), which corresponds to the average Robustness-$\overline{S_r}$ at different sizes of relevant set.

**Robustness-$S_r$**  measures the minimum distortion distance $\epsilon_{S_r}$ when the set of important features $S_r$ are the only region that is perturbable, and the rest of feature values are anchored. Contrary to *Robustness-$\overline{S_r}$*, lower scores on this metric indicate better explanation. We similarly define AUC of Robustness-$S_r$ as the average of Robustness-$\overline{S_r}$ when we vary the size of $|S_r|$.

**Evaluation Procedure**  Note that both Robustness-$\overline{S_r}$ and Robustness-$S_r$ are sensitive to the cardinality of $S_r$. For instance, including all features in $S_r$ will make $\epsilon_{\overline{S_r}}^* = 0$. Therefore, we will only use these criteria to evaluate relevant sets with the same cardinality. For example, to evaluate a feature attribution method that assigns a weight with each feature, we can sort the features by the decending order of weights and then for each set of top-$K$ features with $K = 1, 2, \ldots, d$, we evaluate Robustness-$\overline{S_r}(S_r)$ and plot a curve. A larger (smaller) area under curve indicates a better feature attribution ranking. (See examples in Figure 1).

**Untargeted v.s. Targeted Explanation**  Definition 2.1 corresponds to the untargeted adversarial robustness – a perturbation that changes the predicted class to any label except $y$ is considered as a successful attack. Instead of doing this, our formulation can also extend to **targeted adversarial robustness**, where we replace (1) by

$$\epsilon_{S,t}^* = \{\min_{\boldsymbol{\delta}} \|\boldsymbol{\delta}\|_p \text{ s.t. } f(\boldsymbol{x} + \boldsymbol{\delta}) = t; \boldsymbol{\delta}_{\overline{S}} = 0\}, \tag{2}$$

where $t$ is the targeted class. Using this definition, our approach will try to address the question "Why is this example classified as $y$ instead of $t$", and the important features that optimize this criterion will highlight the contrast between class $y$ and $t$. We will give several interesting results in the experiment section.

**Comparing to existing measurement**  The proposed criteria at the first glance look similar to SSR- and SDR-based measurements. We note that, however, the key differences between our proposed criteria and SSR- (SDR-) based criteria are in two-folds: 1) Conceptually, to measure whether a set of features is important, instead of concerning the prediction change before and after removing the features, we consider whether perturbation on the feature values would significantly alter the prediction. 2) Practically, our proposed criteria allow us to eschew the difficulty of modeling feature removal as discussed in section 1. In fact, as most implementations of removal-based criteria set the values of the features of interest to some fixed reference point, our criteria could be viewed as generalized versions where we consider all possible reference points by allowing perturbations in any directions. As a result, the proposed criteria enjoys a broader view of prediction behavior around the input, and in turn could capture a broader range of important features like the *pertinent negative* features in Dhurandhar et al. (2018), as we shall show in the experiment section.

**Robustness Evaluation under Fixed Anchor Set**  It is known that computing the exact minimum distortion distance in modern neural networks is intractable (Katz et al., 2017), so many different methods have been developed to estimate the value. Adversarial attacks, such as C&W (Carlini & Wagner, 2017) and PGD attack (Madry et al., 2017), aim to find a feasible solution of (1), which leads to an upper bound of $\epsilon_S^*$. They are based on gradient based optimizers which are usually efficient. On the other hand, neural network verification methods aim to provide a lower bound of $\epsilon_S^*$ to ensure that the model prediction will not change within certain perturbation range (Singh et al., 2018; Wong & Kolter, 2018; Weng et al., 2018a; Gehr et al., 2018; Zhang et al., 2018; Wang et al., 2018; Zhang et al., 2019). However, these methods are usually time consuming (often $> 50$ times slower than a backpropagation).

The proposed framework can be combined with any method that aims to approximately compute (1), including attack, verification, and some other statistical estimations. However, for simplicity we only choose to evaluate (1) by the state-of-the-art projected gradient descent (PGD) attack (Madry et al., 2017), since the verification methods are too slow and often lead to much looser estimation as reported in some recent studies (Salman et al., 2019).

## 3  NEW EXPLANATIONS TOWARDS OPTIMIZING THE CRITERIA

Given the new evaluation criteria, a natural follow-up question is how to design explanations that optimize the measurements. Recall that under the proposed criteria the goal of an optimal explanation is to maximize (minimize) robustness-$\overline{S_r}$ (robustness-$S_r$) under the cardinality constraint on $S_r$. Searching for such explanations thus leads to the following optimization problems, (3) for Robustness-$\overline{S_r}$ and (4) for Robustness-$S_r$:

$$\begin{array}{ll} \underset{S_r \in \{0,1\}^d}{\text{maximize}} & g(\boldsymbol{x}, \overline{S_r}) \\ \text{subject to} & \|S_r\|_0 \leq K, \end{array} \quad (3) \qquad \begin{array}{ll} \underset{S_r \in \{0,1\}^d}{\text{minimize}} & g(\boldsymbol{x}, S_r) \\ \text{subject to} & \|S_r\|_0 \leq K, \end{array} \quad (4)$$

where $g(\boldsymbol{x}, S)$ computes the value in Eq. (1), the minimum distortion distance when the features in set $\overline{S_r}$ is not allowed to be perturbed, and $K$ is a pre-defined size constraint on the set $S_r$.

### 3.1  GREEDY ALGORITHM

Directly solving (3) and (4) is challenging since $g$ is an implicit function computed by solving (1) approximately, and furthermore, the discrete input constraint makes it intractible to find the optimal solution. As a result, we propose a greedy-styled algorithm, where we iteratively add the most promising feature into $S_r$ that optimizes the objective at each local step until $S_r$ reaches the size constraint. In other words, we initialize the set $S_r$ as empty, and sequentially solve the following subproblem at every step $t$:

$$\arg\max_i \ g(\boldsymbol{x}, \overline{S_r^t \cup i}), \text{ or } \arg\min_i \ g(\boldsymbol{x}, S_r^t \cup i), \ \forall i \in \overline{S_r} \qquad (5)$$

where $S_r^t$ is the anchor set at step $t$, and $S_r^0 = \emptyset$. We repeat this subprocedure until the size of set $S_r^t$ reaches $K$. We name this method as *Greedy*. A straightforward way for solving (5) is to exhaustively search over every single feature. However, considering a single feature at a time ignores the correlation between features, which tends to introduce noise (see our experimental results). If

| Explanations | Grad | IG | SHAP | LOO | BBMP | Reg-Greedy | Greedy | One-Step Reg |
|---|---|---|---|---|---|---|---|---|
| Robustness-$\overline{S_r}$ | 88.00 | 85.98 | 75.48 | 76.59 | 81.31 | **98.01** | 83.57 | 86.37 |
| Robustness-$S_r$ | 91.72 | 91.97 | 101.49 | 98.82 | 173.90 | **82.81** | 171.56 | 83.59 |

Table 1: Area under curve of the proposed criteria for various explanations on MNIST. The higher the better for Robustness-$\overline{S_r}$; the lower the better for Robustness-$S_r$. Robustness measured with (1).

we consider multiple features at a single step, searching over all possible combinations will become intractable. For example, considering all possible combinations of two features requires $O(d^2)$ evaluations of function $g$ at every step. To consider the joint influence between features efficiently, we propose a smoothed regression version of solving (5) in the following subsection.

## 3.2 Regression Greedy

As considering the correlation between features by searching over all possible subsets of $\overline{S_r^t}$ at every step $t$ is computationally infeasible, we instead propose to approximate the function $g$ by learning a mapping from the binary space of $\{0,1\}^d$, where ones indicate the inclusion of corresponding feature indices and zeros otherwise, to their resulting function value $g(x, \{0,1\}^d)$. Specifically, we can sample a subset $Q \subseteq \{0,1\}^d$ and then consider the following linear regression:

$$\boldsymbol{w}^* = \arg\min_{\boldsymbol{w}} \sum\nolimits_{\boldsymbol{z} \in Q} (\boldsymbol{w}^T \boldsymbol{z} - g(\boldsymbol{x}, \boldsymbol{z}))^2. \tag{6}$$

After the regression is learned, we can treat the coefficients $w$ that correspond to each feature as their approximated effect on the function value of $g$ when they are included into the set $S_r$. By learning such regression where we sample from the possible subsets, we are able to capture the joint relationships between features, and as well smooth out possible noises. In fact, the greedy approach can be viewed as a special case of Reg-Greedy where the sampled subset $Q$, in (6), in each iterative step contains exactly the one-hot encoded vectors with the "on" indices correspond to the remaining feature indices. That is, each one-hot vector indicates the inclusion of a corresponding single feature into the relevant set. In this case, the coefficients of the learned linear regression would be equivalent to the difference in objective value before and after the corresponding feature is included into the relevant set. To take into account feature interactions, Reg-Greedy samples from the whole distribution of $\{0,1\}^d$ where most of the sampled vectors in $Q$ contains multiple "on" indices. In this way, the learned regression captures feature correlations on the objective value and could smooth out possible noises encountered by greedy. There has been a great line of research on studying the interaction between features including the well-known Shapley value which tackles the problem through cooperative game theory perspective. And Lundberg & Lee (2017) proposed a way to use regression with a special kernel to approximate the Shapley value. However, sampling from the whole distribution of $\{0,1\}^d$ could still incur exponential complexity, and using only a reasonable amount of samples might not be able to precisely capture the behavior of the highly non-linear objective function $g$. Therefore, we propose the *Regression Greedy* (Reg-Greedy) approach, where we still run greedy steps to incrementally add indices to $S_r^t$, but at each iteration we run this regression and use the weights to decide which index to be added to $S_r^t$. Note that at each step the samples $Q$ must be in a restricted domain, where indices that are already chosen in $S_r^t$ should be 1 and we sample 0/1 only for the rest of the indices. We distinguish Reg-Greedy from *one-step regression* (One-Step Reg) which directly determines the importance of each feature by merely solving (6) once. By combining regression in a greedy procedure, we are able to gradually narrow down our sampling space (by sampling only from a restricted domain), focusing on the feature interactions between remaining features and the ones that are already added into the relevant set. This enables us to find from the remaining features that have the greatest interaction with the current relevant set, and could in turn maximally optimize the objective value when added into the relevant set. In practice, a sample complexity of $O(d)$ for learning the regression could generally work well.

## 4 Experiments

We present both qualitative and quantitative comparisons in the experiments. For $\|\cdot\|_p$ in (1) and (2), we consider $p = 2$, i.e., the $\ell_2$ norm for all experiments. In quantitative results, including evaluation curves and the corresponding AUC, we report the average over 50 random examples. For

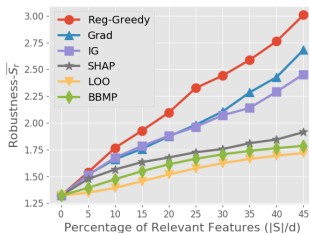 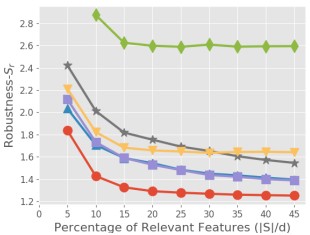 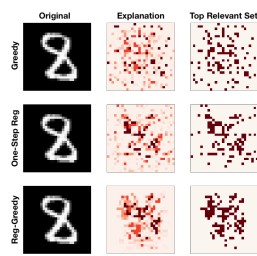

Figure 1: Different Robustness-$\overline{S_r}$ (left) with varying $|\overline{S_r}|$ and Robustness-$S_r$ (right) with varying $|S_r|$. For Robustness-$\overline{S_r}$ (left), the higher the better; for Robustness-$S_r$ (right), the lower the better. We omit points in the plot with value too high to fit in the scale of y-axis.

Figure 2: Visualization on our proposed methods. The top features selected by Reg-Greedy are less noisy.

| Explanations | Grad | IG | SHAP | LOO | BBMP | Reg-Greedy | Greedy | One-Step Reg |
|---|---|---|---|---|---|---|---|---|
| Robustness-$\overline{S_r}$ | 27.13 | 26.01 | 18.25 | 23.54 | 22.60 | **31.62** | 21.16 | 24.54 |
| Robustness-$S_r$ | 45.53 | 46.28 | 60.02 | 52.77 | 154.14 | **43.97** | 58.45 | 47.07 |

Table 2: Area under curve of the proposed criteria for various explanations on ImageNet. The higher the better for Robustness-$\overline{S_r}$; the lower the better for Robustness-$S_r$. Robustness measured with (1).

| Explanations | Grad | IG | SHAP | LOO | BBMP | Reg-Greedy |
|---|---|---|---|---|---|---|
| Insertion | 250.81 | 262.74 | 200.50 | 192.44 | 102.53 | **379.15** |
| Deletion | 281.88 | 273.71 | 362.68 | 442.65 | 527.80 | **159.77** |

Table 3: Area under curve of the Insertion and Deletion criteria for various explanations on MNIST. The higher the better for Insertion; the lower the better for Deletion.

| Explanations | Grad | IG | SHAP | LOO | BBMP | Reg-Greedy |
|---|---|---|---|---|---|---|
| Rank Correlation | 0.3001 | 0.3042 | 0.1108 | 0.4966 | 0.1775 | 0.1835 |

Table 4: Rank correlation between explanations with respect to original and randomized model.

the proposed algorithms, we consider Reg-Greedy (Sec 3.2), One-Step Reg (Sec 3.2) and Greedy (Sec 3.1). For other baselines we include vanilla gradient (Grad) (Shrikumar et al., 2017) and integrated gradient (IG) (Sundararajan et al., 2017) from gradient-based approaches; leave-one-out (LOO), or occlusion-1, (Zeiler & Fergus, 2014; Li et al., 2016) and SHAP (Lundberg & Lee, 2017) from perturbation-based approaches (Ancona et al., 2018), and black-box meaningful perturbation (BBMP) (Fong & Vedaldi, 2017) from SSR/SDR-based approaches. We perform our experiments on two image datasets, MNIST and ImageNet, as well as a text dataset YahooAnswers.

### 4.1 QUANTITATIVE ANALYSIS ACROSS DIFFERENT EXPLANATIONS

**The proposed measurements: Robustness-$\overline{S_r}$ and Robustness-$S_r$.** We compare different explanations under the two proposed criteria, robustness-$\overline{S_r}$ and robustness-$S_r$, and plot their evaluation curves respectively. For ease of comparison, we calculate the area under curve (AUC) for each corresponding evaluation. We list the results in Table 3, and leave the plots in appendix A. As shown in Table 3, under both criteria, comparing to regression-based methods, the pure greedy method usually suffers degraded performances that could be due to the ignorance of feature correlations, which ultimately results in the introduction of noise as shown in Figure 2. Furthermore from the table, we observe that the proposed regression-greedy method consistently outperforms others on both criteria. On one hand, this suggests that the proposed algorithm indeed successfully optimizes towards the criteria; on the other hand, this might indicate the proposed criteria do capture different characteristics of explanations which most of the current explanations do not possess. Another somewhat

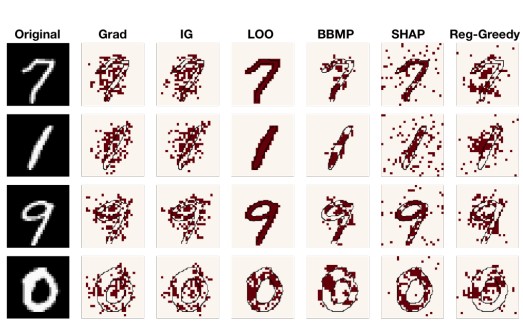

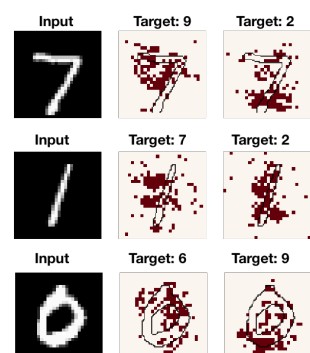

Figure 3: Visualization on top 20 percent relevant features provided by existing explanations.

Figure 4: Visualization of targeted explanation. In each row, we highlight relevant regions explaining why the input is not predicted as the target class.

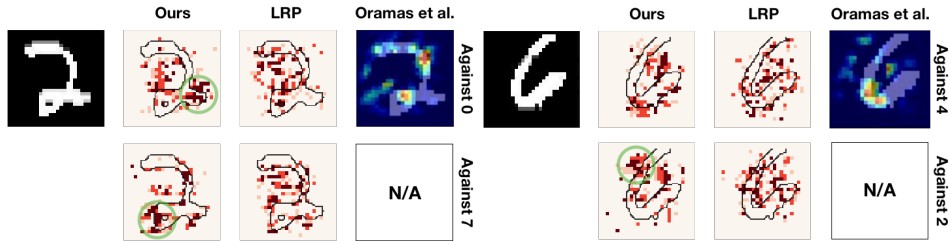

Figure 5: Comparisons between different targeted explanations against different targeted class on MNIST.

interesting finding from the table is that while vanilla gradient has generally been viewed as a baseline method, it nonetheless performs competitively on the proposed criteria. To investigate deeper into such observation, we shall visualize the explanations in the following subsection. For simplicity we will just apply Reg-Greedy with Robustness-$\overline{S_r}$ criterion in the qualitative comparisons with previous methods.

**Existing commonly adopted measurements: Insertion and Deletion.** Indeed, it might not be surprising that Reg-Greedy achieves the best performances on the proposed criteria it is explicitly designed to optimize. To more objectively showcase the usefulness of the proposed explanation, we compare Reg-Greedy with other explanations on existing commonly used quantitative measurements. Particularly, we adopt the *Deletion* and *Insertion* criteria proposed by Petsiuk et al. (2018), which are generalized variants of the *region perturbation* criterion presented in Samek et al. (2016). The Deletion criterion measures the probability drop in the predicted class as top-relevant features, indicated by the given explanation, are progressively removed from the input. On the other hand, the Insertion criterion measures the increase in probability of the predicted class as top-relevant features are gradually revealed from the input whose features are originally all masked. Similar to our proposed criteria, a quick drop (and thus a small area under curve) or a sharp increase (that leads to a large area under curve) in Deletion and Insertion respectively suggest a good explanation as the *selected* top-important features could indeed greatly influence the prediction. In the experiments, we follow Samek et al. (2016) to remove features by setting their values to randomly sampled values. We plot the evaluation curves and report corresponding AUCs in Figure 12 and Table 3. On these additional two criteria, we observe that our proposed method consistently performs favorably against other explanations.

**Sanity Check.** As pointed out in recent literature that an appropriate explanation should at least be loosely related the model being explained (Adebayo et al., 2018), to ensure that our proposed explanation does indeed reflect the model behavior, we conduct the sanity check proposed by (Adebayo et al., NeurIPS'18) to check if our explanations are adequately different when the model parameters are randomly re-initialized. In the experiment, we randomly re-initialize the last fully-connected

layer of the neural network model. We then compute the rank correlation between explanation computed w.r.t. the original model and that w.r.t. the randomized model. From Table 4, we observe that Reg-Greedy has a much lower rank correlation comparing to Grad, IG, and LOO, suggesting that Reg-Greedy is indeed sensitive to model parameter change and is able to pass the sanity check.

## 4.2 QUALITATIVE VISUALIZATIONS

**Visualized Explanations on MNIST.** Figure 3 illustrates the top features identified by various explanation methods. From this figure, we observe that Gradient, IG, SHAP mainly highlights the white pixels in the digit, while gradient and IG are more noisy compared to SHAP. In the contrary, Reg-Greedy focuses on both the "crucial positive" of the digits "pertinent negative" of regions around the digit. For example, in the first row, a 7 might have been predicted as a 4 or 0 if the pixels highlighted by Reg-Greedy are set to 1. Similarly, a 1 may be turned to a 4 or a 7 given additional white pixels to its left, and a 9 may become a 7 if deleted the lower circular part of its head. As a result, Reg-Greedy focuses on *"the region in which perturbing will lead to easier prediction change"*, which includes both the crucial positive pixels and pertinent negative pixels, and provides additional insights that are not captured by the baseline explanations. The superiority of Reg-Greedy is also validated by the better performance on the Robustness-$\overline{S_r}$ score.

**Targeted Explanation.** Recall that in section 2.2, we discussed about the possibility of defining the robustness measurement by considering a targeted distortion distance as formulated in (2). Here, we provide examples, as shown in Figure 4, where we answer the question of *"why the input digit is an A but not a B"* by defining a targeted perturbation distance towards class B as our robustness measurement. In each row of the figure, we provide targeted explanation towards two different target classes for a same input image. Interestingly, as the target classes changes, the generated explanation varies in an interpretatble way. For example, in the first row, we explain why the input digit 7 is not classified as a 9 (middle column) or a 2 (rightmost column). The resulting explanation against 9 highlights the upper-left part of the 7. Semantically, this region is indeed pertinent to the classification between 7 and 9, since turning on the highlighted pixel values in the region (currently black in the original image) will then make the 7 resemble a 9. However, the targeted explanation against 2 highlights a very different but also meaningful region, which is the lower-right part of the 7; since adding a horizontal stroke on the area would turn a 7 into a 2. This finding demonstrates a special characteristic of our explanation which cannot be easily found in most of the existing methods.

While the capability of capturing not only the crucial positive but also the pertinent negative features have also been observed in some recently proposed explanations such as Layer-wise Relevance Propagation (LRP) (Bach et al., 2015), as reported in Samek et al. (2016), as well as the explanation technique proposed in Oramas et al. (2019). Both of the above mentioned methods are not explicitly designed to handle the targeted explanation task which attempt to answer the question "what are the important features that lead to the prediction of class A but not class B", and thus has different limitations. For example, the ability of LRP to capture pertinent negative features in fact heavily depends on the input range. In Samek et al. (2016) where inputs are normalized to have zero mean and a standard deviation of one, the black background will have non-zero value, and LRP would have non-zero attributions on the black background pixels which allows the explanation to capture pertinent negative features. However, as later shown in Dhurandhar et al. (2018), if the input pixel intensity is normalized into the range between 0 and 1 (where background pixels have the values of 0), LRP failed to highlight the pertinent negative pixels, as background would always have zero attribution (since LRP is equivalent to multiplication between Grad and input in a Rectified Linear Unit (ReLU) network as shown in Ancona et al. (2018)). In Oramas et al. (2019), unlike our targeted explanation where we know exactly which targeted class the explanation is suggesting against (and by varying the targeted class we observe varying corresponding explanation given), their method by design does not convey such information. The pertinent negative features highlighted by their method by construction is not directly related to a specific target class, and users in fact need to infer what target class the pertinent negative features are preventing against. To further grasp the difference, we compare our explanation with theirs in Figure 5 (we borrow the results from Oramas et al. (2019) for visualization of their method). Qualitatively, we also observe that our method seems to be giving the most natural explanations. For example, in the first row of left image where the highlighted features are against the class 0, in addition to the left vertical gap (which when presence would make 2 looks like a 0) that is roughly highlighted by all three methods, our method is the

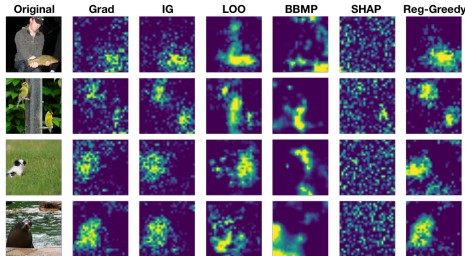

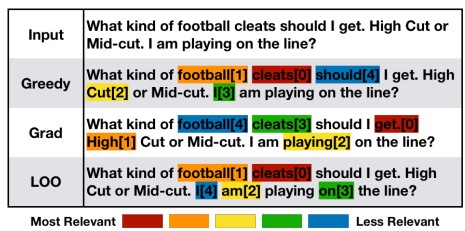

Figure 6: Visualization of different explanations on ImageNet, where the predicted class for each input is "fish", "bird", "dog", and "sea lion".

Figure 7: Explanations on a text classification model where the predicted label for this sentence is "sport".

only one that highlights the right tail part (green circled) of the digit 2 which might also serve as crucial evidence of 2 against 0. Furthermore, as we change the targeted class to 7 (the second row), while LRP seems to be providing similar explanations, we observe that our explanation has a drastic change and highlights the green circled part which when turned off will make 2 becomes a 7. These results might suggest our method is more capable of handling such targeted explanation task.

**Visualized Explanations on ImageNet.** On ImageNet, we as well compare different explanations quantitatively on both of the proposed criteria. We plot the evaluation curves (in appendix A), and compute the corresponding AUC, as listed in Table 2. In general, we observe similar trends as the experiments shown in MNIST. In particular, Reg-Greedy enjoys an overall superior performances than existing explanations on the criteria. In addition, several visualization results in Figure 6 also qualitatively demonstrate that our method provides more compact explanations that focuses more on the actual object being classified.

**Text Classification.** We demonstrate how our explanation method could be applied to text classification models. Note that a length-$n$ sentence is usually represented by $n$ embedding vectors, and thus when applying our Greedy algorithm, at each iteration we will try to add each embedding vector to the set $S_r$ and choose the one with largest reward. Since there are only at most $n$ choices, the Greedy algorithm doesn't suffer much from noise and has similar behavior with Reg-Greedy.

We perform experiments on an LSTM network which learns to classify a given sentence into one of the ten classes (Society, Science, Health, ...). We showcase an example with explanations generated with different methods in Figure 7. We note that although the top-5 relevant keyword sets generated by the three methods do not vary much, the rankings within the highlighted keywords for each explanation are in fact different. We observe that our method Greedy tends to generate explanation that matches human intuition the most. Particularly, to predict the label of "sport", one might consider "cleats", "football", and "cut" as the strongest indications towards the concept "sport".

## 5 RELATED WORK

Our work proposes an objective measurement of feature-based explanation by measuring the "minimum adversarial perturbation" in adversarial literature, which is estimated by adversarial attack. We provide a necessarily incomplete review on related works in objective measurement of explanations and adversarial robustness.

**Objective Measurements for Explanations** Evaluation of explanations has been a difficult problem mainly due to the absence of ground truth (Ancona et al., 2018; Sundararajan et al., 2017). Although one could rely on human intuitions to assess the quality of the generated explanations (Lundberg & Lee, 2017; Doshi-Velez & Kim, 2017), for example, judging whether the explanation focuses on the object of interest in an image classification task, these evaluations subject to human perceptions are prone to fall into the pitfall of favoring user-friendly explanations, such as attributions that visually aligns better with the input image, which might not reflect the model behavior (Adebayo et al., 2018). As a result, in addition to subjective measurements, recent literature has also proposed objective measurements, which is also called functionally-grounded evaluations (Doshi-Velez & Kim, 2017). We roughly categorize existing objective measurements into two families.

This first family of explanation evaluation is called fidelity-based measurement. This includes that *Completeness* or *Sum to Delta* which requires the sum of attributions to equal the prediction difference of the original input and baseline (Sundararajan et al., 2017; Shrikumar et al., 2017); sensitivity-n which further generalizes completeness to any subset of the feature (Ancona et al., 2018); local accuracy Ribeiro et al. (2016); Lundberg & Lee (2017); and *infidelity* which is a framework that encompasses several (Yeh et al., 2019). The general philosophy for this line of methods is to require the sum of attribution value faithfully reflect the change in prediction function value given the presence or absence of certain subset of features. The second family of explanation evaluation are removal-based and preservation-based measurements, which focus on identifying the most important set of features with respect to a particular prediction. The underlying assumption made is that by removing the most (least) salient feature, the resulting function value should drop (increase) the most. (Samek et al., 2016) proposed this idea as an evaluation to evaluate the ranking of feature-attribution score. Later on, Fong & Vedaldi (2017) derive explanations by solving an optimization problem to optimize the evaluation. And Dabkowski & Gal (2017) proposed to learn the explanation generating process by training an auxiliary model.

We note the implicitly in the evaluation process of both fidelity and removal (preservation) based measurement involves computing the change in function value given some set of features being *absent*. However, it is difficult to carefully model the concept of feature absence in practice, as most models by construction are not able to handle inputs with *real* missing features. As a result, previous work has compromised by using approximation to estimate the effect of removing certain features. This includes setting the values of the features to be removed by zero (Ancona et al., 2018; Sundararajan et al., 2017) or the mean value (Lundberg & Lee, 2017), blurred value (Fong & Vedaldi, 2017), random value (Samek et al., 2016; Dabkowski & Gal, 2017), or more advanced generative model that attempts to model the given data distribution (Chang et al., 2018). Unfortunately, such approximations that represent feature absence by setting the their values to some predefined distribution would inevitably introduce bias into the evaluation process. With the presence of this inherent caveat, we are thus inspired to adopt another angle to tackle the explanation problem.

**Adversarial Robustness**  Adversarial robustness has been extensively studied in the past few years. The adversarial robustness of a machine learning model on a given sample can be defined as the shortest distance from the sample to the decision boundary, which corresponds to our definition in (1). Algorithms have been proposed for finding adversarial examples (feasible solutions of (1)), including (Goodfellow et al., 2014; Carlini & Wagner, 2017; Madry et al., 2017). However, those algorithms only work for neural networks, while for other models such as tree based models or nearest neighbor classifiers, adversarial examples can be found by decision based attacks (Brendel et al., 2017; Cheng et al., 2018; Chen et al., 2019). Therefore the proposed framework can also be used in other decision based classifiers. On the other hand, several works aim to solve the neural network verification problem, which is equivalent to finding a lower bound of (1). Examples include (Singh et al., 2018; Wong & Kolter, 2018; Zhang et al., 2018). In principal, our work can also apply these verification methods for getting an approximate solution of (1), but in practice they are very slow to run and often gives loose lower bounds on regular trained networks.

Our work is also closely related to related works that consider the question "For situation A, why was the outcome B and not C", which we call counterfactual explanations. Xu et al. (2018) add group sparsity regularization to adversarial attack to enforce semantic structure for the perturbation, which is more interpretable. Ribeiro et al. (2018) find a set of features that once fixed, probability of the prediction is high when perturbing other features. Goyal et al. (2019) show how one could change the input feature such that the system would output a different class, where the change is limited to replacing a part of input feature by a part of an distractor image. Dhurandhar et al. (2018) consider the pertinent negative in a binary setting by solving a carefully designed loss function.

## 6  CONCLUSION

In this paper, we establish the link between a set of features to a prediction with a new evaluation criteria, robustness analysis, which measures the minimum tolerance of adversarial perturbation. Furthermore, we develop a new explanation method to find important set of features to optimize this new criterion. Experimental results demonstrate that the proposed new explanations are indeed capturing significant feature sets across multiple domains.

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

# A    EVALUATION CURVES

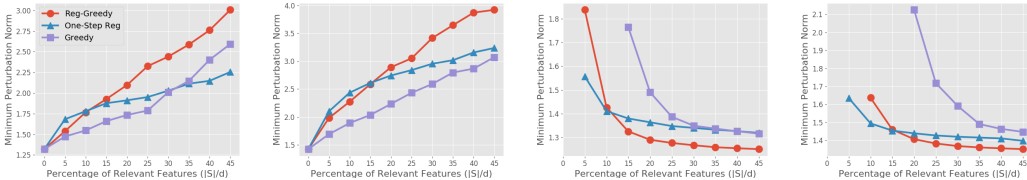

Figure 8: Comparisons between our proposed methods under different criteria. From left to right: untargeted Robustness-$\overline{S_r}$, targeted Robustness-$\overline{S_r}$, untargeted Robustness-$S_r$, targeted Robustness-$S_r$. We omit points in the plot with value too high to fit in the scale of y-axis.

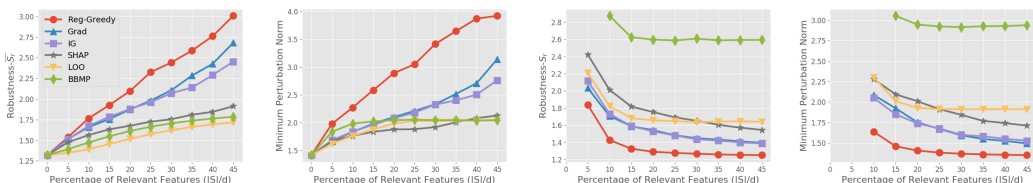

Figure 9: Comparisons between our proposed methods and existing explanations under different criteria. From left to right: untargeted Robustness-$\overline{S_r}$, targeted Robustness-$\overline{S_r}$, untargeted Robustness-$S_r$, targeted Robustness-$S_r$. We omit points in the plot with value too high to fit in the scale of y-axis.

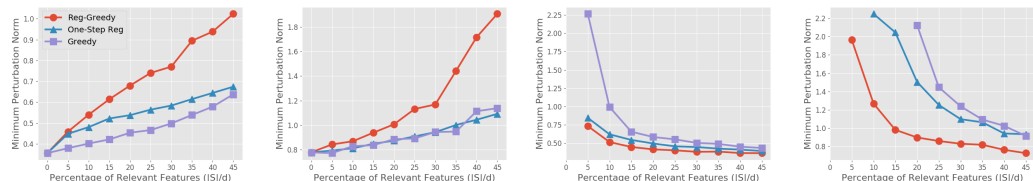

Figure 10: Comparisons between our proposed methods under different criteria on ImageNet. From left to right: untargeted Robustness-$\overline{S_r}$, targeted Robustness-$\overline{S_r}$, untargeted Robustness-$S_r$, targeted Robustness-$S_r$. We omit points in the plot with value too high to fit in the scale of y-axis.

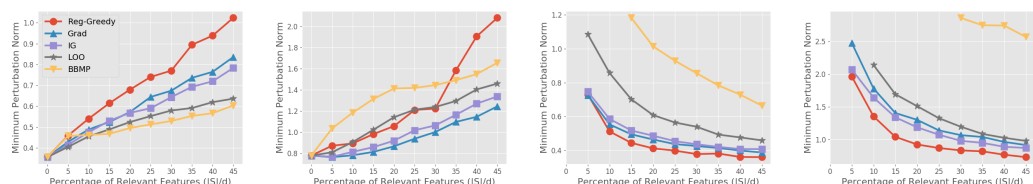

Figure 11: Comparisons between our proposed methods under different criteria on ImageNet. From left to right: untargeted Robustness-$\overline{S_r}$, targeted Robustness-$\overline{S_r}$, untargeted Robustness-$S_r$, targeted Robustness-$S_r$. We omit points in the plot with value too high to fit in the scale of y-axis.

# B    EVALUATION ON EXISTING OBJECTIVE MEASUREMENTS

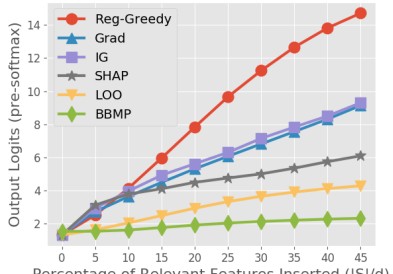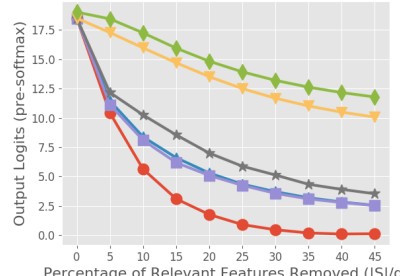

Figure 12: Comparisons between explanations under different criteria on MNIST. Left figure: change in output logits as relevant features are inserted into the input. Right figure: change in output logits as relevant features are removed from the input.

| Explanations | Grad | IG | SHAP | LOO | BBMP | Reg-Greedy |
|---|---|---|---|---|---|---|
| Insertion | 250.81 | 262.74 | 200.50 | 192.44 | 102.53 | **379.15** |
| Deletion | 281.88 | 273.71 | 362.68 | 442.65 | 527.80 | **159.77** |

Table 5: Area under curve of the Insertion and Deletion criteria for various explanations on MNIST. The higher the better for Insertion; the lower the better for Deletion.

# C    T-TEST ON AUC

| Explanations | Grad | IG | SHAP | LOO | BBMP |
|---|---|---|---|---|---|
| Robustness-$\overline{S_r}$ | win | win | win | win | win |
| Robustness-$S_r$ | win | win | win | win | win |

Table 6: The proposed Reg-Greedy versus other explanations on MNIST under our proposed criteria with Student's $t$-test at 95% confidence level.

| Explanations | Grad | IG | SHAP | LOO | BBMP |
|---|---|---|---|---|---|
| Insertion | win | win | win | win | win |
| Deletion | win | win | win | win | win |

Table 7: The proposed Reg-Greedy versus other explanations on MNIST under Insertion and Deletion criteria with Student's $t$-test at 95% confidence level.

# D    SANITY CHECK

| Explanations | Grad | IG | SHAP | LOO | BBMP | Reg-Greedy |
|---|---|---|---|---|---|---|
| Rank Correlation | 0.3001 | 0.3042 | 0.1108 | 0.4966 | 0.1775 | 0.1835 |

Table 8: Rank correlation between explanations with respect to original and randomized model.

# E COMPARISONS ON TARGETED EXPLANATION

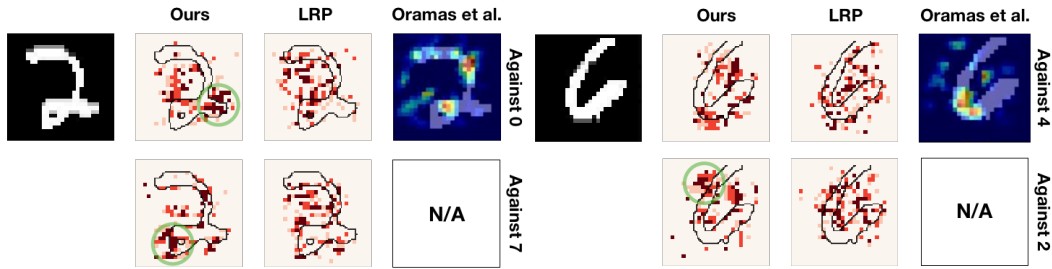

Figure 13: Comparisons between different targeted explanations against different targeted class on MNIST.

# F HEATMAP VISUALIZATION ON MNIST

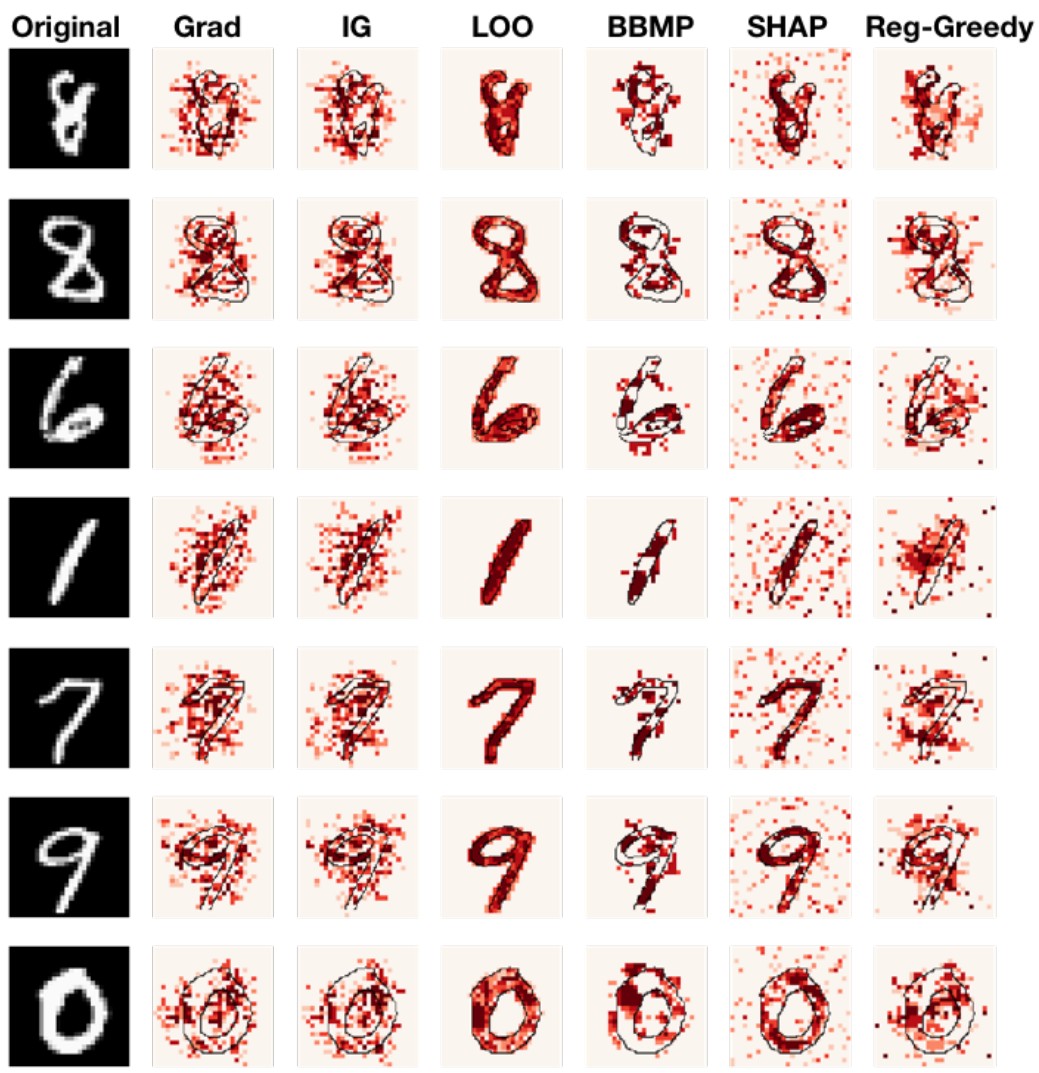

Figure 14: Heatmap Visualization of different explanations on MNIST.

