# OpenReview forum: "Evaluations and Methods for Explanation through Robustness Analysis"
_ICLR.cc/2020/Conference — Reject_

### Official Review · AnonReviewer1 · 2019-10-23
**Official Blind Review #1**

**Rating:** 6

**Review:**



The manuscript proposes a method for model explanation and two metrics for the evaluation of methods for model explanation based on robustness analysis. More specifically, two complementary, yet very related, criteria are proposed: i) robustness to perturbations on irrelevant features and ii) robustness to perturbations in relevant features. Moreover, different from existing works which defined the perturbation values following different somewhat-fixed procedures, the proposed method aims at allowing perturbations in any directions.

A greedy algorithm optimizing these criteria is proposed in order to produce a method able to highlight important features of the input and justify/explain model predictions.
In addition, the proposed robustness criteria are used as metrics to assess the performance of methods for model explanation.

Experiments on models addressing image classification and text classification, shows the performance of the proposed method w.r.t. existing work.

The manuscript has a good flow, and its content is easy to follow. The proposed method is sound, well motivated, and very well founded. The formal presentation of the proposed method is good. I appreciate the fact that evaluation covers different modalities of data, i.e. text and images.

My main criticism over the manuscript is the following:

In Sec. 3.3, when describing the first criterion, it is stated that the size |S_r|, i.e, the amount of anchors, could be defined by the used. In my opinion, this may not be applicable since in theory the amount of relevant/irrelevant features is unknown before hand. In that case the proposed AUC-based method seems more adequate. Could you comment on this?


In  Sec. 3, a pre-defined size K is introduced. Later in Sec. 3.1 it is stated that the greedy algorithm uses this size as a stopping criterion for the optimization of the proposed robustness criteria. Could you indicate how this size K is defined in practice? Is there a principled way to define it? What is the effect of this parameter on the performance of the proposed method? An ablation study focused on this parameter would provide further insights into the inner workings of the proposed method and would improve the manuscript.


In Sec.4 it is stated that only 50 random examples are considered when reporting results. When comparing the performance across the different methods, are these random 50 examples fixed or always re-sampled? Also, given the size of the considered datasets, where the number of images in their test sets is in the order of the thousands, it is hard to grasp how representative are the reported results?

In the same paragraph discussed above, it is mentioned that the GRAD method performs competitively on the proposed criteria. It might be interesting to further positioning the proposed method w.r.t. GRAD. Given the comparable performance achieved by GRAD and its relative simplicity, it would be hard to motivate why not choose GRAD instead of the proposed method? Could you provide some discussion on this?

In Sec.4 (pag. 6) it is stated the the proposed regression-greedy method outperforms other methods in these criteria. In my opinion this trend shouldn't be surprising given the fact that the proposed method is specifically optimized on such criteria as it is clearly stated by the title of Sec.3.

Fig.3 and Fig.4 display binary images indicating the top-n features selected by different methods. Perhaps it would be more informative to have a heatmap highlighting/grading the entire input space. This may throw more light on the performance of the compared methods.

In Adebayo et al., NIPS'18 (and very related efforts), there are presented a set of sanity checks to be applied to explanation methods to ensure their predictions are relate to the class and model being predicted. Could you provide any indication on whether the proposed method passes these checks?

In Sec.4 (visualization) it is stated that the proposed method effectively highlights crucial positive pixels as well as pertinent negative pixels. A similar capability has also being reported earlier in Samek et al., Trans NNLS'16 and Oramas et al. ICLR'19. Since the visualization analysis (discussed in pag.6) focuses exclusively on this capability. There should be a comparison between the proposed method and the two mentioned works.



**Experience Assessment:**

I have published one or two papers in this area.

**Review Assessment: Checking Correctness Of Derivations And Theory:**

I carefully checked the derivations and theory.

**Review Assessment: Checking Correctness Of Experiments:**

I carefully checked the experiments.

**Review Assessment: Thoroughness In Paper Reading:**

I read the paper thoroughly.

---

> ### Author Response · Authors · 2019-11-12
> **Response to Reviewer #1**
>
> Thank you for your careful review and your helpful comments. We have incorporated your suggestions into our revision. We address the questions raised in the response below.
>
> Q1: The amount of relevant/irrelevant features is unknown beforehand. In that case the proposed AUC-based method seems more adequate. Could you comment on this? Could you indicate how this size K is defined in practice? Is there a principled way to define it? What is the effect of this parameter on the performance of the proposed method?
>
> A1: One general goal of feature-based explanations is to extract a "compact" set of relevant features for a given model prediction, since the most straightforward yet vacuous explanation is simply highlighting all features as relevant (which does not constitute a meaningful explanation). However, because the number of true relevant features is in general unknown beforehand (as Reviewer #1 notes), the predominant approach recent papers have considered is to output the top-K important features, for varying values for K. For example, in attribution methods such as Grad and IG, we could take the top-K features with the highest attribution scores. And K is usually set to varying values so that we generate relevant feature set explanations of different sizes. Similarly, in our proposed method, we allow users to set the value of K such that our explanation could identify the top-K most important features to the prediction. In our experiments, we vary the value of K such that our explanations provide sets of relevant features of sizes 5%, 10%, ..., 50% of the total number of features. Then, for each of these relevant sets with differing sizes, we could apply the proposed evaluation criteria to evaluate their quality, which yields a single evaluation curve shown in Figure 1.
>
> Such evaluation curves measure the quality of an explanation by considering differing sizes of relevant features, and the AUC then reflects the overall quality of the explanation. In the case where users have no knowledge about the number of relevant features, our paper thus suggests the use of AUC of the evaluation curve, which as the reviewer notes is indeed more than adequate as an evaluation. But to also provide a rationale for evaluations of differing values of K: it provides the quality of the relevant sets along multiple points on the evaluation curve, instead of a single numerical summary. And in some special use cases, the users might indeed be interested in a pre-defined size of relevant set, e.g. top-20% relevant features. But as the reviewer suggests, in our paper, we do recommend the use of AUC, which we also use to compare across different explanations, in addition to plotting the whole evaluation curves to illustrate the performances of different explanations at various sizes of relevant set.
>
> Q2: When comparing the performance across the different methods, are these random 50 examples fixed or always re-sampled? Also, given the size of the considered datasets, where the number of images in their test sets is in the order of the thousands, it is hard to grasp how representative are the reported results?
>
> A2: In the revision, we have increased the number of testing examples from 50 to 100. In the experiments, we first randomly sample 100 examples from the entire test set. We then compare different explanation methods on the same set of 100 examples. We find that the relative performance of explanations is not sensitive to the number of testing examples being changed from 50 to 100. To further justify that the proposed method indeed enjoys a better performance on different criteria (our proposed criteria and other existing criteria which we shall introduce later), we conduct pairwise Student's t-test comparing the performances of our proposed method and other existing explanations. As shown in Appendix C, our method does enjoy a statistically significantly better performance over other explanations across various criteria.

---

> > ### Author Response · Authors · 2019-11-12
> > **Response to Reviewer #1 (Cont.)**
> >
> >
> > Q3: Given the comparable performance achieved by GRAD and its relative simplicity, it would be hard to motivate why not choose GRAD instead of the proposed method? Could you provide some discussion on this?
> >
> > A3:  While we do observe that Grad has somewhat competitive performance on the proposed criteria, it has also been known that Grad has different shortcomings as well. One such caveat is its notorious "saturation" problem. For example, consider a binary classification model that takes two-variable input $f(x_1, x_2) = sign(1 - ReLU(1 - x_1) - 0.5)$ (inspired by  Sundararajan et al., ICML'17). Naturally, one would regard $x_1$ as relevant and $x_2$ as irrelevant since the classification result in fact only depends on $x_1$. However, for a given input $(x_1, x_2) = (2, 5)$, the explanation (attribution score) provided by Grad will be $(0, 0)$ (as the function becomes flat at $x_1 = 1$), which fails to distinguish between relevant and irrelevant features. In this case, our proposed method could still distinguish between the relevance level of $x_1$ and $x_2$, as the Robustness-$S_r$ will be $1.5$ if $S_r = \{x_1\}$; and infinity if $S_r = \{x_2\}$. The main difference (and perhaps also advantage) of our method over Grad is that our method explains the model behavior from a more global viewpoint, as opposed to Grad which only consider the function sensitivity to each individual feature locally.
> >
> >
> > Q4: It is stated the the proposed regression-greedy method outperforms other methods in these criteria. In my opinion this trend shouldn't be surprising given the fact that the proposed method is specifically optimized on such criteria as it is clearly stated by the title of Sec.3.
> >
> > A4: We agree that it is not too surprising the proposed method has the best performance on the criteria it is explicitly designed to optimize; though it serves as a sanity check of our method. To more comprehensively showcase the usefulness of the proposed method, we conduct an additional set of experiments comparing our method with other explanations on various existing commonly adopted quantitative measurements in the literature. In particular, we adopt the Deletion and Insertion criteria (Petsiuk et al., BMVC'18) which are generalized variants of the region perturbation criterion (Samek et al., Trans NNLS'16). The deletion criterion measures the drop in the probability of a class as top-relevant features (given by the explanation) are progressively removed from the input. Similar to our proposed criterion Robustness-$S_r$, a quick drop, and thus a small area under the curve, suggests a good explanation as the "selected" relevant features indeed greatly influence the prediction. On the other hand, the insertion criterion measures the increase in the probability of a class as top-relevant features are gradually introduced into the input. In the experiments, we follow (Samek et al., Trans NNLS'16) to remove features by setting their values to randomly sampled values. We plot the evaluation curves and report corresponding AUCs in Appendix B. On these additional two criteria, we observe that our proposed method consistently performs favorably against other explanations. The t-test results in Table 7 (Appendix C) also indicate such outperformance is indeed significant.
> >
> > Q5: Perhaps it would be more informative to have a heatmap highlighting/grading the entire input space.
> >
> > A5: In the revision, we have included more visualization results with heatmaps indicating the relative importance of different features in Appendix F.

---

> > > ### Author Response · Authors · 2019-11-12
> > > **Response to Reviewer #1 (Cont.)**
> > >
> > >
> > > Q6: Could you provide any indication on whether the proposed method passes these sanity checks?
> > >
> > > A6: To ensure that our proposed explanation does indeed reflect the model behavior, we conduct the sanity check proposed by (Adebayo et al., NeurIPS'18) to check if our explanations look different when the model parameters being explained are randomly re-initialized. In the experiment, we randomly re-initialize the last fully-connected layer of the neural network. We then compute the rank correlation between explanation computed w.r.t. the original model and that w.r.t. the randomized model. From Table 8 in Appendix D, we observe that our method has a much lower rank correlation comparing to Grad, IG, and LOO, suggesting that our method is indeed sensitive to model parameter change and is able to pass the sanity check.
> > >
> > > Q7: A similar capability (finding both crucial positive pixels as well as pertinent negative pixels) has also being reported earlier in Samek et al., Trans NNLS'16 and Oramas et al. ICLR'19. Since the visualization analysis (discussed in pag.6) focuses exclusively on this capability. There should be a comparison between the proposed method and the two mentioned works.
> > >
> > > A7: In Samek et al., they showed that Layer-wise Relevance Propagation (LRP) (Bach et al., PLOS ONE) has the capability of capturing both crucial positive pixels as well as pertinent negative pixels. This capability of LRP in fact depends on the input range where inputs are normalized to have zero mean and a standard deviation of one in Samek et al. In this case, the black background will have non-zero value, and LRP would have non-zero attributions on the black background pixels which allows the explanation to capture pertinent negative features. However, as later on shown in Dhurandhar et al. (NeurIPS'18), if the input scale is in the range of [0, 1] (where background pixels have the values of 0), LRP failed to highlight the pertinent negative pixels, as background would always have the zero attribution (since LRP is equivalent to Grad * Input in a ReLu network as shown in Ancona et al., ICLR'18).
> > > In Oramas et al. ICLR'19, a similar capability of highlighting the pertinent negative features has also been reported. Although their explanation method could capture both pertinent positive as well as pertinent negative that supports the prediction, their method is not explicitly designed for answering the question of "what are the important features that leads to the prediction of A but not B''. In other words, unlike our method where we can specify different B and observe different explanations given, their method cannot handle such requests by design. Users in fact need to infer what target class the pertinent negative features are suggesting against. Finally, we compare our explanation with theirs in Appendix E Figure 13 (we borrow the results from Oramas et al. for visualization of their method). Qualitatively, we also observe that our method seems to be giving the most natural explanations. For example, in the first row of left image where the highlighted features are against the class 0, in addition to the left vertical gap (which when presence would make 2 looks like a 0) that is roughly highlighted by all three methods, our method is the only one that highlights the right tail part (green circled) of the digit 2 which might also serve as crucial evidence of 2 against 0. Furthermore, as we change the targeted class to 7 (the second row), while LRP seems to be providing similar explanations, we observe that our explanation has a drastic change and highlights the green circled part which when turned off will make 2 becomes a 7. These results might suggest our method is more capable of handling such targeted explanation task.

---

### Official Review · AnonReviewer4 · 2019-10-31
**Official Blind Review #4**

**Rating:** 3

**Review:**

SUMMARY

The authors propose an intuitive new measure (definition 2.1) of feature importance based on a robustness criterion (with two variants of equations 3 and 4). Two optimisers are proposed for finding the most important features according to this measure, in order to explain why a classifier is making a certain prediction.

The experiments are both qualitative (showing pixel-wise importance maps for image problems) and quantitative (showing how well the various feature importance scoring algorithms do albeit on the measure explicitly optimised by the new proposed algorithms).

COMMENTS

The paper organisation is clear enough though the English needs a little work to make it read really nicely.

The proposed measures are natural and intuitive. Especially the Robustness-$\hat{S_r}$ is an interesting twist on the more obvious Robustness-$S_r$.

While the measures are interesting, the justification for them is somewhat weak. This amounts to

1) Quantitative experiments which seem to only test how well each method works in relation to the very metric which only your proposed method directly optimises - this is nice but not surprising. Also, the way you define the AUC of your measure seems a little strange. I don't know why different baselines appear in different comparisons as in e.g. tables 1 and 2. Finally your curves in appendix A don't seem to have the same number of points for each method in all cases?

2) Qualitative examples with images and text. These are nice, but alas only qualitative. Also, it seems as though not all baselines are included in all examples (even figures 3 and 5, which are analogous, include different baselines).

It seems like the paper needs either 1) a quantitative evaluation that is not subjective. Surely, the evaluation metric should not match the (novel) objective of the proposed method? Or, 2) a theoretical result in support of the new measures.

The Reg-Greedy algorithm is a major contribution of this paper, but receives very little explanation. Indeed, perhaps the clearest quantitative statement of the paper is that Reg-Greedy beats Greedy. Is this a common method for optimising w.r.t. a subset? Is it similar to other methods? I felt that Reg-Greedy is a really nice idea but the paper did not do it justice.

DETAILS

Perhaps unifying (3) and (4) by defining a single g that subsumes both cases would be neater.

Please define g in equation (1) rather than in words after equation (4).

It should be S_r in the subscripts of (3) and (4)

"Crutial" spelling

FINALLY

I'm open to be swayed on any of the above points, pending the author feedback.


**Experience Assessment:**

I have published one or two papers in this area.

**Review Assessment: Checking Correctness Of Derivations And Theory:**

N/A

**Review Assessment: Checking Correctness Of Experiments:**

I assessed the sensibility of the experiments.

**Review Assessment: Thoroughness In Paper Reading:**

I read the paper at least twice and used my best judgement in assessing the paper.

---

> ### Author Response · Authors · 2019-11-12
> **Response to Reviewer #4**
>
> Thank you for your careful review and your helpful comments. We have incorporated your suggestions into our revision and addressed the questions raised as follows.
>
> Q1: Quantitative experiments which seem to only test how well each method works in relation to the very metric which only your proposed method directly optimises - this is nice but not surprising.
>
> A1: We agree that it is not too surprising the proposed method has the best performance on the criteria it is explicitly designed to optimize; though it serves as a sanity check of our method. To more objectively showcase the usefulness of the proposed method, we conduct an additional set of experiments comparing our method with other explanations on varied existing commonly adopted quantitative measurements in the literature. In particular, we adopt the Deletion and Insertion criteria (Petsiuk et al., BMVC'18) which are generalized variants of the region perturbation criterion (Samek et al., Trans NNLS'16).  The deletion criterion measures the drop in the probability of a class as top-relevant features (given by the explanation) are progressively removed from the input. Similar to our proposed criterion Robustness-$S_r$, a quick drop, and thus a small area under the curve, suggests a good explanation as the "selected" relevant features indeed greatly influence the prediction. On the other hand, the insertion criterion measures the increase in the probability of a class as top-relevant features are gradually introduced into the input. In the experiments, we follow (Samek et al., Trans NNLS'16) to remove features by setting their values to randomly sampled values. We plot the evaluation curves and report corresponding AUCs in Appendix B. On these additional two criteria, we observe that our proposed method consistently performs favorably against other explanations. The t-test results in Table 7 (Appendix C) also indicate such outperformance is indeed significant.
>
> Q2: Also, the way you define the AUC of your measure seems a little strange.
>
> A2: Our definition of AUC simply measures the area under evaluation curves as the ones shown in Figure 1. Since for all explanations, we evaluate their Robustness-$\overline{S_r}$/Robustness-$S_r$ on the same set of varying sizes of relevant set (from 0% to 45% of the total number of features), the AUC is in fact proportional to the average Robustness-$\overline{S_r}$/Robustness-$S_r$ evaluated at different points along the x-axis. The AUC reflects the overall explanation quality by assuming different amount of underlying true relevant features. We have clarified the definition of AUC in the revision.
>
> Q3: Your curves in appendix A don't seem to have the same number of points for each method in all cases?
>
> A3: For the figures in Appendix A, the reason why some curves seem to have less points than others is because we omit the points whose value is too high to fit into the scale of y-axis in the plot. We have clarified this point in the revision.
>
> Q4: I don't know why different baselines appear in different comparisons as in e.g. tables 1 and 2. It seems as though not all baselines are included in all examples (even figures 3 and 5 (figure 6 in current revision), which are analogous, include different baselines).
>
> A4: In Table 2, we originally omit the results for SHAP on ImageNet since it has known to be computationally prohibiting on high-dimensional images. In the revision, we have included its results on ImageNet by implementing 8 by 8 super-pixel to reduce the dimension of feature space. We have included both of its quantitative and qualitative results in Table 2 and Figure 6 respectively. In addition, for Figure 3 and Figure 6, we have included consistent set of baselines for the qualitative results.
>
> Q5: Qualitative examples with images and text. These are nice, but alas only qualitative.
>
> A5: We agree that qualitative results cannot serve as the only measurement to evaluate explanations. As a result, we use different quantitative criteria to complement the findings from the qualitative visualizations, and hope both objective and subjective results together could bring more insights into different explanation strategies.

---

> > ### Author Response · Authors · 2019-11-12
> > **Response to Reviewer #4 (Cont.)**
> >
> >
> > Q6: The Reg-Greedy algorithm is a major contribution of this paper, but receives very little explanation. Indeed, perhaps the clearest quantitative statement of the paper is that Reg-Greedy beats Greedy. Is this a common method for optimising w.r.t. a subset? Is it similar to other methods?
> >
> > A6: Thank you for the suggestion. We agree that Reg-Greedy is indeed a very interesting approach for optimizing w.r.t. a subset. To the best of our knowledge, we have not noticed this method being used in the literature, but we can note the following connections to several different families of methods. First of all, as discussed in the paper, one main motivation of using regression over the greedy approach is that pure greedy only considers individual influence of a single feature on the objective function separately for each feature. This short-term, or one-step look ahead, can actually be very noisy and thus taking such greedy path along the way may not benefit the long-term optimization goal. By ignoring the interaction between features, greedy can easily fail to capture the set of features that together might have the greatest influence on the objective function.
> > In fact, the greedy approach can be viewed as a special case of Reg-Greedy where the sampled subset $Q$ (in Eq. 6) in each iterative step contains exactly the one-hot encoded vectors with the "on" indices correspond to the remaining feature indices. That is, each one-hot vector indicates the inclusion of a corresponding single feature into the relevant set. In this case, the coefficients of the learned linear regression would be equivalent to the difference in objective value before and after the corresponding feature is included into the relevant set. To take into account feature interactions, Reg-Greedy samples from the whole distribution of $\{0, 1\}^d$ where most of the sampled vectors in $Q$ contains multiple "on" indices. In this way, the learned regression captures feature correlations on the objective value and could smooth out possible noises encountered by greedy. In fact, there has been a great line of research on studying the interaction between features including the well-known Shapley value which tackles the problem through cooperative game theory perspective. And (Lundberg and Lee, NIPS 2017) proposed a way to use regression with a special kernel to approximate the Shapley value. However, sampling from the whole distribution of $\{0, 1\}^d $ could still incur exponential complexity, and using only a reasonable amount of samples might not be able to precisely capture the behavior of the highly non-linear objective function. As a result, by combining regression in a greedy procedure, we are able to gradually narrow down our sampling space (by sampling only from a restricted domain), focusing on the feature interactions between remaining features and the ones that are already added into the relevant set. This enables us to find from the remaining features that have the greatest interaction with the current relevant set, and could in turn maximally optimize the objective value when added into the relevant set. The iterative approach thus gives the main advantage of Reg-Greedy over one-step regression. We have enriched our discussion on Reg-greedy in the revision.
> >
> > Q7: Overall polishing of the paper
> >
> > A7: We have revised the typos and incorporated your suggestions on paper writing to make the paper neater and easier to read. We will conduct another round of paper polishing in the revision.

---

### Public Comment · ~TING_TING_SUN1 · 2019-10-09
**Some questions and a related paper**

Hi, this is a great work but I have some questions.

In this paper, you assume that "the model could tolerate a larger degree of perturbation on the less important and non-anchored features". I wonder is there any work supports this assumption?  Since robustness and importance are two concepts. Some important features can be quite robust to perturbation. Maybe it is more reasonable to evaluate the importance of features based on the theory of the Shapley value.

Besides, I think this paper may be related to your work:
Zhang X , Wang N , Shen H , et al. Interpretable Deep Learning under Fire. 2018. arXiv:1812.00891

---

> ### Author Response · Authors · 2019-10-09
> **Our assumptions fall in line with existing evaluation measurements**
>
> Hi,
>
> Thank you for your comment.
>
> We note that the assumption "the model could tolerate a larger degree of perturbation on the less important and non-anchored features" is analogous to the common assumption "the model prediction does not change much when the less important features are removed", which is adopted in several popular existing explanation evaluations. However, as we discuss in the paper (section 1 and section 5), the notion of feature removal is generally difficult to model and is often implemented by setting the feature value to zero or some random value in practice. Such removal practice inevitably is prone to introduce bias into the evaluation process. As a result, we instead consider the idea of prediction robustness as allowing perturbation is a much more general notion that does not introduce extra bias but with similar underlying meaning.
>
> Specifically, the assumptions "the model prediction does not change much when the less important features are removed" and "the model prediction change more when the more important features are removed" are considered by SSR- and SDR-based evaluations [1], which later on became commonly adopted in the literature [2, 3]. Moreover, such assumptions also lie in fidelity-based attribution evaluations implicitly, since fidelity-based evaluations assume that the feature importance should be related to the average performance drop of the model when the feature is removed. A previous work [4] has shown that Shapley value for explanation also optimizes such a fidelity measurement with a specific perturbation prior.
>
> Therefore, we believe that our assumption falls in line with the majority of explanation evaluations but additionally resolves the caveat of removing features which introduces bias to the evaluation process (and thus the explanation). We agree that robustness and importance are two concepts, but we believe that they are related.
>
> We have read the suggested related work which considers more on the problem of sensitivity of explanations to adversarial inputs. While we believe it is not directly related to our work, we will surely consider enriching our related work section to further include this line of studies.
>
> [1] Evaluating the visualization of what a deep neural network has learned. Wojciech Samek, Alexander Binder, Grégoire Montavon, Sebastian Lapuschkin, and Klaus-Robert Müller. IEEE transactions on neural networks and learning systems, 28(11):2660–2673, 2016.
> [2] RISE: Randomized Input Sampling for Explanation of Black-box Models. Vitali Petsiuk, Abir Das, Kate Saenko. BMVC 2018.
> [3] Explaining image classifiers by counterfactual generation. Chun-Hao Chang, Elliot Creager, Andrew A. Goldenberg, and David Kristjanson Duvenaud. ICLR 2019.
> [4] On the (in)fidelity and sensitivity for explanations. Chih-Kuan Yeh, Cheng-Yu Hsieh, Arun Sai Suggala, David I. Inouye, and Pradeep Ravikumar. Arxiv 2019.

---

### Author Response · Authors · 2019-11-12
**General Response**

We thank all reviewers for their constructive and helpful comments. We have incorporated their suggestions into our revision, and have hopefully addressed all of their questions raised. In brief: (a) we conduct an additional set of experiments evaluating our proposed method via a suite of existing commonly adopted quantitative measurements, (b) we ran a Student's t-test to positively verify the statistical significance of the performance improvements of our proposed method over existing explanations on various criteria, (c) we conducted a sanity check based on model parameter randomization and verified that our proposed explanation does indeed pass the test, and finally, (d) we significantly enriched the comparisons, discussions, and make clarifications in our manuscript.

---

### Decision · Program_Chairs · 2019-12-19

**Decision:**

Reject

**Comment:**

The paper proposes an approach for finding an explainable subset of features by choosing features that simultaneously are: most important for the prediction task, and robust against adversarial perturbation. The paper provides quantitative and qualitative evidence that the proposed method works.

The paper had two reviews (both borderline), and the while the authors responded enthusiastically, the reviewers did not further engage during the discussion period.

The paper has a promising idea, but the presentation and execution in its current form have been found to be not convincing by the reviewers. Unfortunately, the submission as it stands is not yet suitable for ICLR.